# Study of Damage Prediction of Carbon Fiber Tows Using Eddy Current Measurement

**DOI:** 10.3390/polym15204182

**Published:** 2023-10-21

**Authors:** Jeong U Roh, Hyein Kwon, Sangjin Lee, Jae Chung Ha, Seong Baek Yang, Haeseong Lee, Dong-Jun Kwon

**Affiliations:** 1Composites Convergence Research Center, Korea Textile Machinery Convergence Research Institute, Gyeongsan 38542, Republic of Korea; juroh@kotmi.re.kr (J.U.R.); hikwon@kotmi.re.kr (H.K.); 2High-Tech Product Research Headquarters, Korea Textile Machinery Convergence Research Institute, Gyeongsan 38542, Republic of Korea; sjlee@kotmi.re.kr; 3Vastek Korea Co., Ltd., Bucheon 14502, Republic of Korea; 4Research Institute for Green Energy Convergence Technology, Gyeongsang National University, Jinju 52828, Republic of Korea; sbyang@gnu.ac.kr; 5Department of Carbon & Nanomaterials Engineering, Jeonju University, Jeonju 55069, Republic of Korea; haeseong@ju.ac.kr; 6Department of Materials Science and Convergence Technology, Gyeongsang National University, Jinju 52828, Republic of Korea

**Keywords:** carbon fiber, damage detection, Eddy current testing, impedance, evaluation

## Abstract

When manufacturing fiber-reinforced composites, it is possible to improve the quality of fiber steel fire and reduce the number of cracks in the finished product if it is possible to quickly identify the defects of the fiber tow. Therefore, in this study, we developed a method to identify the condition of carbon fiber tow using eddy current test (ECT), which is used to improve the quality of composite materials. Using the eddy current detection sensor, we checked the impedance results according to the condition of the CF tow. We found that the materials of the workbench used in the experiment greatly affected the ECT results, so it is necessary to use a material with a non-conductive and smooth surface. We evaluated the impedance results of the carbon fiber at 2 mm intervals using the ECT sensor and summarized the impedance results according to the fiber width direction, presenting the condition of the section as a constant of variation (CV). If the condition of the carbon fiber tow was unstable, the deviation of the CV per section was large. In particular, the deviation of the CV per section was more than 0.15 when the arrangement of the fibers was changed, foreign substances were formed on the surface of the fibers, and damage occurred in the direction of the fiber width of more than 4 mm, so it was easy to evaluate the quality on CF tow.

## 1. Introduction

As carbon fiber-reinforced plastics (CFRP) gain popularity in transportation and structures, their long-term reliability focuses on enhancing. In general, metal-based structural materials have thus far been characterized by non-destructive testing [1,2]. However, since CFRPs are subject to variations in composition, suitable non-destructive evaluation methods need to be found and criteria for defects and analytical methods need to be established.

The most commonly used method for non-destructive testing of CFRP is ultrasonic testing [3]. Ultrasonic testing (UT) is a non-destructive method used to inspect materials, including composites, for defects [4]. It operates by generating high-frequency sound waves using a transducer, typically equipped with a piezoelectric crystal that converts electrical energy into mechanical vibrations. These sound waves are directed into the material, penetrating its layers and encountering any internal imperfections such as interfaces, voids, delamination, or cracks. As the ultrasonic waves interact with these defects, some are reflected back to the transducer, while others continue to propagate. By measuring the time it takes for waves to travel to the defect and return, technicians can determine the depth and size of the flaws [5]. Flaws such as voids, delamination, cracks, or fiber misalignment are identified and characterized through this analysis [6,7]. The collected data is displayed on an oscilloscope or digitally recorded for further examination. Technicians interpret this data to assess the composite material’s condition and determine if it complies with specified quality and safety standards.

While UT techniques can provide a highly-sensitive and non-destructive assessment of the health of a structure, they do have some drawbacks [8,9,10]. Composites are often anisotropic, meaning their mechanical properties can vary in different directions [11]. The direction of ultrasonic waves and their interaction with composite layers can complicate the interpretation of results. Composite structures can have complex shapes and curvatures, which can affect the propagation and reception of ultrasonic waves. Specialized transducers and techniques may be needed to address these challenges. The surface finish of composite components can impact the quality of UT inspections [12]. Interpreting UT data for composites requires specialized training and expertise. Technicians must be skilled in recognizing and characterizing the unique features and indications that can be present in composite materials. High-quality UT equipment can be expensive, and specialized transducers and probes may be required for inspecting composite materials, adding to the overall cost. UT may have limitations in inspecting very thick composite structures due to the attenuation of ultrasonic waves as they pass through the material [13,14].

To ensure effective transmission of ultrasonic waves into the material, a coupling medium (e.g., gel, water) is often required. Proper application of this medium is critical for accurate inspections. To address these challenges, ongoing research and development efforts are focused on improving UT techniques for composites. This includes the development of advanced transducers, signal processing algorithms, and modeling approaches to enhance the accuracy and reliability of inspections [15].

Although the accuracy of UT is high, it is necessary to perform non-destructive evaluation (NDE) of composite materials with non-contact methods to increase the ease of evaluation [16]. In addition, it is necessary to make it easy to investigate the results obtained by NDE to increase the utilization of NDE for composite materials. For these reasons, Eddy current detection is currently being studied [17].

Eddy current testing (ECT) relies on the principle of electromagnetic induction. When an alternating current (AC) passes through a coil or probe, it generates a changing magnetic field [18]. When this coil or probe is brought close to a conductive material (such as a metal), Eddy currents are induced in the material. These Eddy currents generate their own magnetic fields, which interact with the original field created by the coil. The interaction between these fields produces electrical impedance changes in the coil, which can be measured and analyzed. The resulting impedance differences in the specimen’s condition can be used to detect damage, cracks, deformation, and more in composites.

ECT can be used to measure the thickness of conductive coatings on composite surfaces [19]. For example, if a composite material is coated with a thin layer of conductive material (e.g., for electromagnetic interference shielding or corrosion protection), ECT can be used to assess the thickness and uniformity of the coating [20,21]. In some composite structures, conductive inserts or inclusions may be present, such as metal fasteners or embedded sensors. ECT can help detect these conductive components and assess their condition within the composite.

The Eddy current test method is considered to be highly useful for condition identification of thin materials such as thin carbon fiber reinforced composites, carbon fiber tows, prepregs, and tow prepregs. In particular, the use of carbon fiber tows, prepregs, CF tapes, etc., is increasing due to the increasing use of hydrogen containers and pultruded products [22,23,24]. If stable intermediate materials are not used in these fields, the stability of the structure will be greatly reduced. Therefore, a technique that enables non-contact and high-speed evaluation of the condition of basic materials such as CF tow in a continuous process is required. Eddy current measurement is applicable to a wide range of materials with conductivity. However, it is greatly influenced by the thickness of the sample. In particular, for composite materials, while non-destructive evaluation is not possible for glass fiber composites, it is feasible for carbon fiber-reinforced composites that possess conductivity.

CF spread tow, that was measured using ECT in this study, is manufactured by spreading the fiber tow in the width direction, and has the advantage of reducing the weight per unit area and the resin impregnation distance. In addition, woven fabric using the spread tow can reduce the resin rich zone inside the composite material due to the thin tow thickness. Furthermore, studies have shown that applying the fabric to composite materials results in limiting cracks under fatigue load and improving fatigue life [25,26,27,28,29]. In terms of product value, carbon fiber spread tow can improve product value due to its beautiful appearance and can obtain various mechanical properties introduced above, therefore it is mainly used as a surface layer of composite materials. However, in the spreading process of CF, damage to the fiber often occurs, and sometimes the fibers are not spread evenly, causing them to clump together or widen. Therefore, the need for quality control through uniformity inspection of spread tow is emerging at manufacturing sites.

In the case of carbon fibers, a conductive material in the form of fibers, it is important to manage the arrangement of the fibers because the degree to which the filament strands are arranged can cause performance degradation in localized areas. Optical image analysis is a common method for checking the homogeneity of filaments in carbon fibers and can be used to analyze the arrangement of filaments on the surface, but it has limitations in analyzing the presence and density of filaments arranged in the thickness direction. A method for continuously measuring the homogeneity of carbon fibers with large surface areas is Eddy current measurement, which utilizes electrical conductivity properties. An Edy current measurement is used to detect discontinuities in materials that conduct electricity.

In this study, we investigated the Eddy current detection method to evaluate the homogeneity of CF tows. Differences in impedance results depending on the state of the CF tow were analyzed. Impedance was measured in the row and column directions at 2 mm intervals on a CF tow with a width of 20 mm and a length of 100 mm, and the coefficient of variation (CV) results were used to classify the state of the tow. The signal analysis was checked for differences in the workbench holding the specimen. Ultimately, we established an evaluation criteria method to identify the condition of CF tow, identified the most problematic CF tow defect types by Eddy current detection method, and presented an evaluation method to organize impedance results. We analyzed the tensile results of individual carbon fiber filaments that constitute CF tow to assess the accuracy of classifying CF tow defects based on ECT. Consequently, we observed a proportional relationship between the variation in tensile strength of individual filaments due to CF tow damage and the defect state identified by ECT. In other words, we were able to determine the state of the CF tow using the defect information obtained from ECT analysis.

## 2. Materials and Methods

### 2.1. Sample Preparation

Carbon fiber (no: T700SC-12000-60E) was obtained from Toray Industries, Inc. (Seoul, Republic of Korea). The spread tow with a width of 20 mm was prepared using tow spreading equipment (Harmony Industry Co., Ltd., Tokyo, Japan) based on air spreading. The spread tow was fixed in shape using a PA binder (model no: PR SK-1) procured from Tokyo Printing Ink Mfg. Co., Ltd. (Tokyo, Japan). It had an average particle diameter of 70 μm. Comparison samples were prepared as follows for representative defects that occured during the manufacturing stage of spread tow. CF tows based on damage status can be classified as follows; First, there is a case of fibers within 1 mm in width and intra-fiber defects present (assessed as normal), and secondly, there is a case of fibers larger than 1 mm in width and inter-fiber defects present (poor). Moreover, the presence of dust, contact by foreign objects on the fiber tow, and separation of fiber and inter-fiber interfaces may occur rarely in the process. Meanwhile, to confirm the resolution of ECT, samples were prepared by applying artificial damage. Samples were prepared by reflecting the fiber fractured in the horizontal direction and the vertical direction on the spread tow. Through this, the difference in impedance results was compared according to the damage direction in a unidirectional fiber specimen (anisotropic material).

### 2.2. Eddy Current Testing (ECT) System

A 150 mm × 150 mm square workbench was 3D printed using acrylonitrile butadiene styrene (ABS). The specimen was fixed on the workbench. As shown in Figure 1, we analyzed the difference between using copper plate, glass, and ABS on the workbench.

To minimize the lift-off phenomenon that occurs when utilizing Eddy-current detection sensors, the samples were fixed with PVA solution and dried before testing. The probe (ECS-03, Megacoil, São Carlos, SP, Brazil) was mounted on a KUKA 6-axis robot arm (KR 6 R900 sixx, KUKA, Seoul, Republic of Korea) to continuously measure the impedance of the CF tow and precisely control its position. The condition of the CF tow was measured at 2 mm intervals, and the impedance results for each section were measured about the CF tow specimen (size: 20 mm × 100 mm). The data were organized into sections of 10 impedance results section per length (width: 20 mm, measured at 2 mm intervals, total 10 sections; length: 100 mm, measured at 2 mm intervals, total 50 sections). Therefore, the results expressing the condition of the specimen were evaluated using 50 coefficient of variation (CV) results in the length direction. The CV is a representative statistical measure expressed as a percentile by multiplying the value obtained by dividing the standard deviation by the mean by 100. A lower CV indicates a stable result with a small deviation, while a higher CV signifies a larger deviation. The probe used for the measurements had a separate excitation coil that generated Eddy currents and a receiver coil that detected magnetic field signals, allowing high currents to flow. The specification of the half transmission probe is 1 mm radius and 27 MHz frequency. It is known that the longitudinal electrical conductivity ($\sigma_{L}$) of unidirectional single layered carbon fiber is $5\times 10^{6}\;S/m$, and the transverse electrical conductivity ($\sigma_{T}$) is $1\times 10^{3}\;S/m$, which may vary depending on the spacing between the filaments according to the fiber volume fraction, the type of carbon fiber, etc. Therefore, the CF tow specimens used in this experiment, with a thickness of about 300 μm, were accurately evaluated. A tool-path for the robot was created to ensure that the probe measurement position was the same for all samples, and that only the area where the sample was present was measured whenever possible. The Eddy current detection system was configured and used as shown in Figure 2. Utilizing the impedance results confirmed by ECT, we calculated CV values for different segments. Based on this, we categorized the extent of damage in CF tow into four major stages. In cases of fiber breakage, we further classified the damage into fiber breakage in the fiber’s length direction and fiber rupture in the width direction. Additionally, we observed variations in cracks and ECT results based on the extent of separation between fibers and interfaces between fibers.

### 2.3. Single Fiber Tensile Test

To characterize the state of the CF tow as evaluated by ECT, we conducted tensile testing on individual carbon fiber filaments using the ASTM D-3379 standard method [30,31]. The procedure for the tensile testing followed the methods outlined in the reference [32]. We used a paper frame to secure the individual fibers for the tensile experiment. Based on the ECT analysis, we classified the cracks in the CF tow into four major types: normal CF tow, CF tow with separation at the fiber interface, CF tow with foreign substances on the surface, and CF tow with observed fiber breakage. Considering these significant variables, we prepared ten specimens for the tensile testing of individual fibers. We conducted the experiments using the LD10 model from Lloid (AMETEK, Inc., Berwyn, PA, USA), maintaining a testing speed of 1 mm/min. By utilizing the tensile results of individual fibers based on the classification state of the CF tow, we elucidated the state of the CF tow classified by ECT.

## 3. Results and Discussion

### 3.1. Workbench Affects Eddy Current Detection

The results obtained from measuring the workbench using a CV sensing sensor in Figure 3 have been summarized as follows. In the case of the copper plate, its conductivity due to copper led to differences in CV values caused by variations in surface or internal composition [33,34]. As shown in the figure, the difference between the maximum CV value and the minimum CV value was significantly larger compared to other workbench materials. ABS material also exhibited the second-largest CV difference. Most of the CV results were around 0.1, with the glass plate showing the lowest CV variation. However, in the case of ABS, a polymer material, it could potentially share similar characteristics with the glass plate. Still, surface roughness. generated as a result of 3D printing manufacturing, caused variations in CV values across different sections. Therefore, it was confirmed that using a non-conductive workbench, rather than a conductive one, could enhance the objectivity of evaluating the state of the CF tow.

### 3.2. Eddy Current Detection Results Based on Deformation/Damage Morphology of the CF Tow

As shown in Figure 4, when conducting evaluations using a copper plate as the workbench, the overall CV results consistently exceeded 0.4. If relatively uniform CV results are achieved, it can be explained that the condition of CF tow is consistent across segments. However, when measured using a copper plate, significant variations in CV were observed, even in cases of a stable tow condition, as seen in Figure 4a, where CV results exhibited a quadratic trend. In cases where clear cracks were present, as depicted in Figure 4b, it was observed that the CV average values in those segments increased sharply compared to segment-wise CV averages. This was attributed to the impedance difference caused by clear damage to the CF tow layers. However, it was challenging to detect cracks with a size of less than 4 mm in both width and length. This suggests that using a copper plate as the workbench may have led to higher baseline CV results.

CF tow with minimal damage, as shown in Figure 5a, exhibits relatively low CV results. Although some fiber sections have bends that slightly increase the CV values, overall stability in the CF tow formation with stable inter-fiber interfaces results in consistent CV readings. As seen in Figure 3, under ABS workbench conditions, impedance results for different segments typically fall within 0.1 or lower. Therefore, a condition similar to Figure 5a signifies a stable CF tow.

However, when the fibers were cut lengthwise, as seen in Figure 5b, the width of the cut segments becomes crucial. The typical configuration of CF tow includes fiber-to-fiber contact. However, damage that leads to the cutting of CFs and impairment of inter-fiber interfaces results in increased CV compared to the stable CV of CF tow. In other words, if segment-wise CV values are compiled, and their average shows an upward trend compared to the baseline, it indicates damage to the CF tow.

As seen in Figure 5c, where fibers are damaged lengthwise, the width of damage slightly increases. However, as observed in Figure 4b, when the damaged width remains within 4 mm, significant CV changes are not observed. Figure 5d represents situations where damage occurs in two ways: damage along the length of fibers and cutting of fibers along their width. In such cases, overall higher CV values are noted. In essence, damage to the width of the fiber greater than 4 mm leads to significant CV changes. Furthermore, CF tow with damage tends to produce higher CV values compared to the average CV values of undamaged or stable ABS. These trends in increased CV can be used to assess the extent of damage to CF tow [35].

Overall, we could not observe a significant increase in CV due to fiber damage in the fiber length direction of the CF tow. The reason for this is that even if fibers are fractured in the fiber alignment direction, ECT indicates detection signals at the interface between fibers rather than when fibers are fractured. However, as shown in Figure 5d, when fiber breakage occurs in the fiber width direction or when cracks of more than 4 mm occur in the fiber width direction, we were able to detect an increased signal compared to the conventional CV range. In other words, we confirmed that the utilization of ECT for detecting fiber cracks in CF tow is possible depending on the size of cracks within the fibers. Furthermore, we confirmed that if fiber damage occurs in the fiber length direction, there is minimal change in impedance CV values in ECT measurements.

In the current results of the impedance measurement on CF tow, observations reveal a bent CF tow, as depicted in Figure 6. When only bent sections of CF are present, as shown in Figure 6a, the overall CV results exhibit a two-dimensional trend. Furthermore, in cases like Figure 6b, where both the curvature of fibers and the separation between fibers and fiber interfaces are observed, there is relatively minimal alteration in the CV results. This is because the effect of inter-fiber interface separation and the curvature of CF overlap lead to impedance changes. However, these changes are very subtle, with CV value variations within 0.3. To detect such subtle changes precisely, further research and enhancement of sensor performance are required. As seen in Figure 5a–c, detecting damage to the CF tow using ECT requires a high level of precision when there is no change in crack or fiber alignment of 4 mm or more in the fiber width direction. Furthermore, it was observed that the deformation of the fiber width direction and fiber alignment of the CF tow could result in the most significant differences in impedance observed through the CV in the ECT technique.

Figure 7 presents the evaluation results of CF tow on a glass working surface. Since glass is a non-conductive material, it provides a low-impedance working environment. Consequently, the overall CV results for CF tow exhibited low variation. Similarly, CF tow with minimal damage, as seen in Figure 7a, showed a low rate of change and less curvature in the trend line.

However, when significant separation occurred between fibers in CF tow, as shown in Figure 7b, unstable regions were observed, resulting in pronounced curvature in the overall CV trend line. In cases where fiber damage was minor, such as Figure 7c, there was a slight CV change due to the damage, but the magnitude of change was minimal. Figure 7d demonstrates that when fiber discontinuity occurred with a separation of 4 mm or more, significant changes in the CV signal were observed. In summary, while lengthwise fiber variations can be detected through the overall CV trend line, the degree of change is relatively small. On the other hand, when damage occurs in the fiber width direction and exceeds 4 mm, significant changes in segmental CV can be observed on the graph. Furthermore, these results can be related to the findings from Figure 5d. Figure 5d represents fibers with a width-wise breakage of more than 4 mm and a relatively large length of 10 mm. On the other hand, in the case of Figure 7d, the fibers had a width-wise breakage of more than 4 mm but were weaker in the fiber length direction. This suggests that the extent of damage in the fiber width direction influences the CV results in ECT signals. Damage in the fiber length direction may affect the number of interfacial cracks between fibers, but due to the greater influence of the fiber alignment angle on ECT, evaluating the fiber arrangement in CF tow using ECT proves to be the most effective method.

In summary, the observations from Figure 5d highlight the impact of damage in the fiber width direction, especially when it exceeds 4 mm, on the ECT signal’s CV results. Additionally, Figure 7d further emphasizes that damage extending over 4 mm in the fiber width direction, coupled with lower strength in the fiber length direction, affects the CV results in ECT. This underscores the significance of evaluating fiber arrangement in CF tow using ECT, as it is influenced more by the fiber alignment angle than the number of interfacial cracks between fibers resulting from damage in the fiber length direction.

Figure 8 presents an analysis of the extent of damage observed in CF tow using the Eddy current probing method. While previous data focused on changes related to the damage to CF tow itself, Figure 8 emphasizes the impact of foreign substances adhered to CF tow as indicated by CV values. In the cases of Figure 8a,b, significant CV increases were observed in specific sections due to CF debris adhered to the CF tow surface, and the extent of this change could be predicted. Figure 8c was performed on a glass working surface and showed relatively lower CV values compared to Figure 8a,b, but it still exhibited changes in CV results due to foreign substances. In other words, it was confirmed that foreign substances within CF tow could be detected using the Eddy current probing method. To distinguish the impact of CF arrangement, Figure 8d presents results from measuring CF veil, which showed a relatively consistent CV trend. Based on these results, it was concluded that detecting laminations by individual CF fibers using the Eddy current sensing method may be challenging. However, it can readily detect cases where some parts of CF tow are contaminated with foreign substances and exist on the pure CF tow surface [36,37,38].

Figure 9 summarizes the potential states of CF tow that can ultimately occur. In most cases, conditions similar to Figure 9a,b can be considered as undamaged and high-quality CF tow. Additionally, it was observed that detecting cases where the gap between fibers is within 4 mm using the current sensing method is challenging. When the CF tow is torn, as shown in Figure 9c, changes in CV results can be detected, and this can be observed through an overall increase in CV values. In the case of Figure 9d, using the current sensing method provided the most accurate detection. This scenario represents the most critical situation, and detecting it using the current sensing method is considered a significant discovery. For Figure 9e,f, it was anticipated that significant variations in current would occur due to fiber breakage. However, it was confirmed that substantial changes are not observed when the fiber is cut in the width direction by less than 4 mm.

In summary, Figure 9a shows uniformly arranged CF tow, indicating a stable fiber-to-fiber interface within the CF tow. Figure 9b illustrates fiber separation at the 1mm level when observed visually. Although this condition does not display severe cracks, void formation during actual application could lead to reduced mechanical properties. Figure 9c–f represent forms of CF tow that can potentially create significant cracks in composite materials. In the case of Figure 9c, interfaces between fibers in the CF tow are dispersed, indicating a defective state. However, Figure 4, Figure 5, Figure 6, Figure 7 and Figure 8 confirm limitations in detecting fiber cracks using the Eddy current measurement method. Hardware improvements to enhance sensor sensitivity are necessary when utilizing the Eddy current measurement method. Figure 9c,d represent cases where CF tow damage is distinctly detected. Through the Eddy current measurement method, impedance values of the conductive CF tow can be detected, and significant changes in the impedance coefficient are observed when foreign substances adhere or external CF tow is attached. Damage to the CF tow can be detected, and such damage due to array changes is detectable through the Eddy current measurement method. Finally, in the case of Figure 9e,f, the Eddy current measurement method did not detect damage locations when there was damage in the fiber width or length direction within 4mm. Since the eddy current measurement method can identify CF tow damage and the presence of foreign substances, it is advisable to apply non-destructive evaluation towards detecting macroscopic fiber arrays rather than microscopic cracks.

The CF tow can be categorized into four major categories, as seen in Figure 9: (a) normal condition similar to standard carbon fibers, (b) presence of cracks in the fiber interface, such as in Figure 9b,c, (c) fibers with foreign substances adhered or embedded on the fiber surface, as shown in Figure 9d, and (d) cases where some fibers within the CF tow are cut or severed, depicted in Figure 9e,f. In the case of normal CF tow (Figure 9a), the CV results were relatively consistent. When there was separation between fibers or fiber-to-fiber interface damage (Figure 9b,c), the CV results showed an increase ranging from 0.1 to 0.2 compared to the surrounding regions. Depending on the degree of non-uniformity in the CF tow, the results were observed in a two-dimensional curve. In areas where the CF tow had bends or pronounced curvatures, the trend line of the two-dimensional curve showed larger curvatures. In cases where foreign substances were present on the fiber surface, fiber damage was observed. This showed the most significant change in CV results, allowing for measurement with ECT sensors at specific locations. Finally, fiber breakage was observed in cases where cracks in the width direction of CF tow were 4 mm or more, and the length of the fibers in the fiber length direction had minimal impact on the ECT signal changes.

As shown in Figure 10, we characterized the state of the CF tow through single-fiber tensile experiments [39,40]. In the case of normal CF tow, similar to Figure 9a, most of the tensile strengths of the individual fibers were uniform. However, CF tow under conditions like Figure 9b,c showed an uneven state in the fiber results, as depicted in Figure 10b. Figure 10c represents the evaluation of CF tow when foreign substances were present on the fibers, similar to Figure 9d. It was observed that the presence of foreign substances significantly altered the fiber strength and led to a considerable variation in tensile results. This difference is attributed to the impairment of fiber homogeneity due to foreign substances on the fibers and the formation of cracks on the fiber surface. Finally, in the case of Figure 10d, the result of the single-fiber experiment was obtained using a portion of CF tow where fiber cutting was observed. It was confirmed that the tensile strength of the individual fiber was greatly degraded compared to the normal state. In this way, we were able to classify the state of the CF tow into four categories using ECT based on the results of single-fiber tensile experiments under various CF tow conditions.

## 4. Conclusions

This study proposes a method to assess the condition of CF tow by utilizing the Eddy current sensing technique and summarizing the impedance results in the fiber width direction as a CV. When conducting Eddy current sensing, it is suitable to use a working surface with uniform properties, and segment-wise organization of impedance results represented as CV allowed for the prediction of CF tow condition. While direct detection of fiber-to-fiber interface separation within the CF tow was challenging using the Eddy current sensing method, it was possible to confirm it through the curvature of the trend line and the increase in average CV values from the impedance results. Detecting fiber breakage directly through Eddy current sensing was difficult; however, when fiber width-wise cuts of 4 mm or more occurred, changes in CV values could be observed. Lastly, in cases where foreign substances were present on the CF tow, the most significant CV changes were observed, and the extent of the foreign substance’s presence could be determined. Based on these results, it is possible to assess the condition of CF tow through the Eddy current sensing method. The application of such evaluation systems and analysis methods in real-time processes is expected to drive advancements in composite material manufacturing technology.

## Figures and Tables

**Figure 1 polymers-15-04182-f001:**
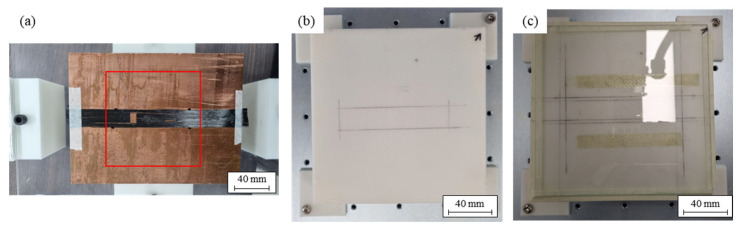
Specimen fixtures with the different material: (**a**) copper plate; (**b**) ABS plate; and (**c**) glass plate.

**Figure 2 polymers-15-04182-f002:**
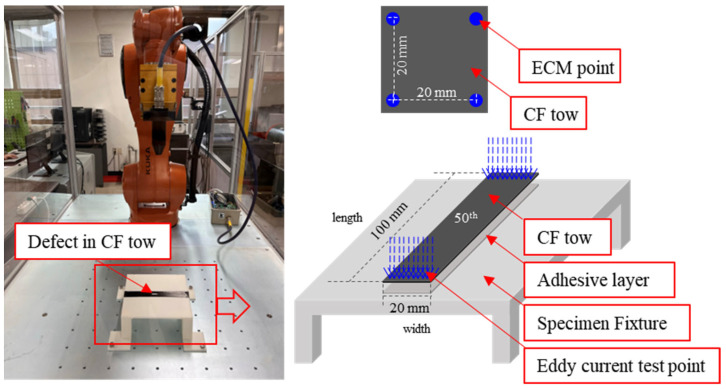
Photograph of evaluating CF tows using the Eddy current probing detection method in the lab.

**Figure 3 polymers-15-04182-f003:**
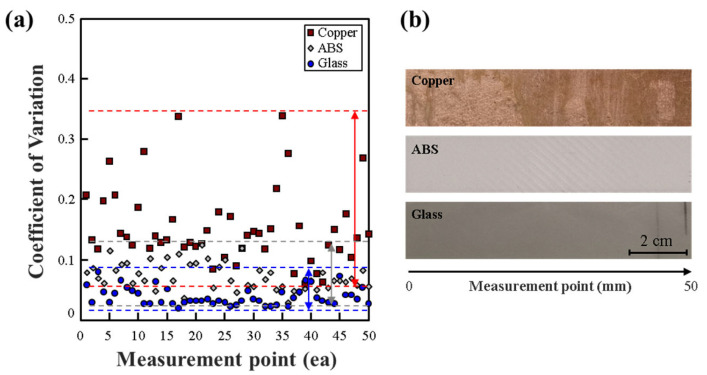
(**a**) Coefficient of variation (CV) for the impedance of a workbench with different materials. (**b**) Photographical images and measurement point of workbenches with various materials copper, ABS, glass.

**Figure 4 polymers-15-04182-f004:**
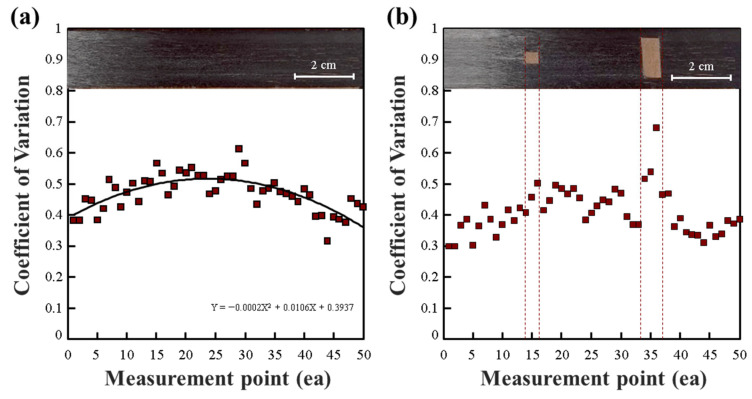
CV for the impedance of a CF tow measured on a copper workbench: (**a**) normal CF tow, (**b**) damaged CF tow.

**Figure 5 polymers-15-04182-f005:**
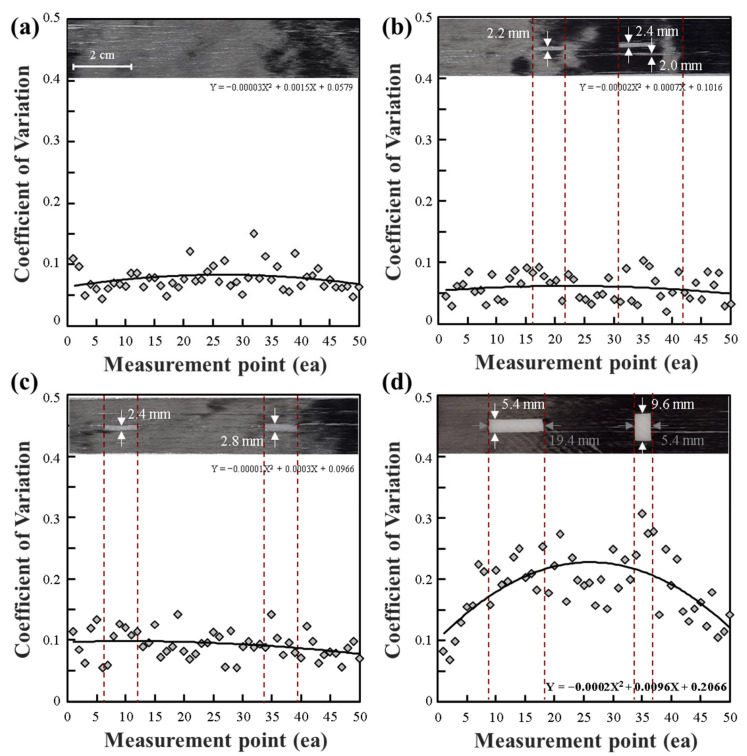
CV for the impedance of a CF tow measured on an ABS workbench: (**a**) normal CF tow, (**b**–**d**) damaged CF tow.

**Figure 6 polymers-15-04182-f006:**
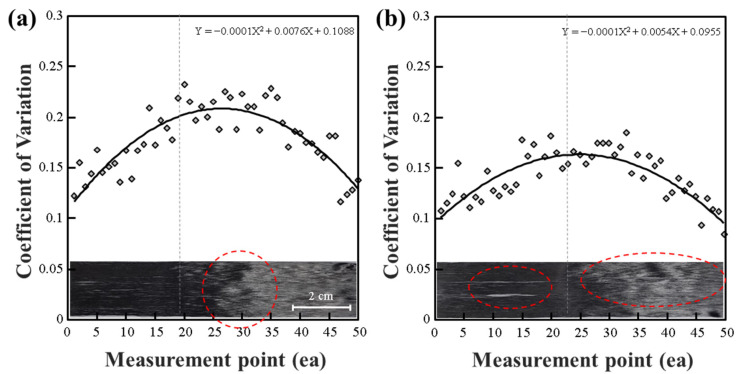
CV for the impedance of a CF tow measured on an ABS workbench: (**a**) curved CF tow; (**b**) curved and interface damaged CF tow.

**Figure 7 polymers-15-04182-f007:**
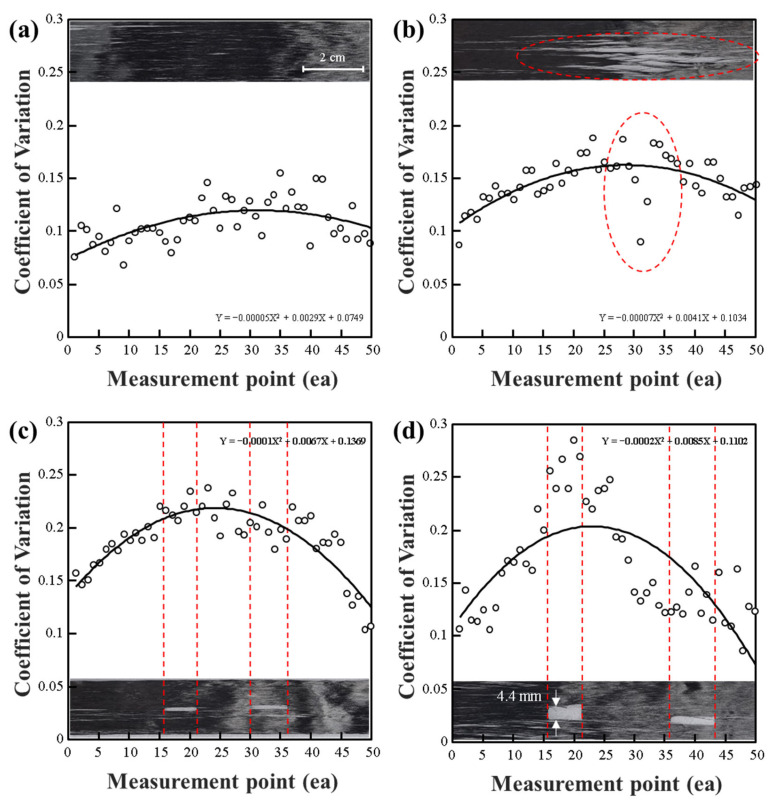
CV for the impedance of a CF tow measured on a glass workbench: (**a**) normal; (**b**) the CF tow is interface damaged in the fiber length direction; (**c**) the fiber is partially cut in the fiber length direction; and (**d**) the CF tow is largely damaged.

**Figure 8 polymers-15-04182-f008:**
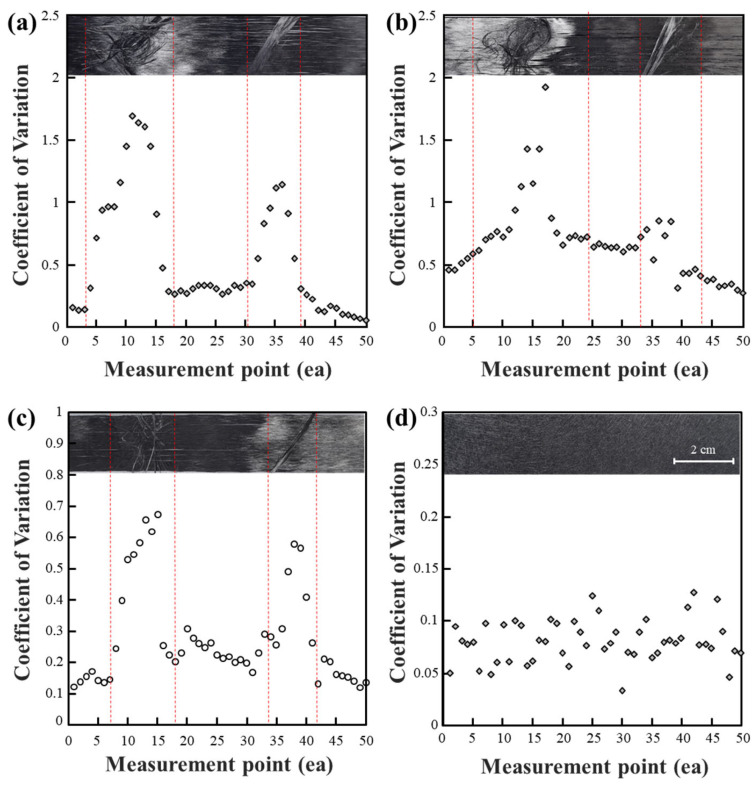
(**a**,**b**) CV for the impedance of a CF tow with debris measured on an ABS; (**c**) CV for the impedance of a CF tow with debris measured on a glass; (**d**) CV for the impedance of a CF veil measured on an ABS.

**Figure 9 polymers-15-04182-f009:**
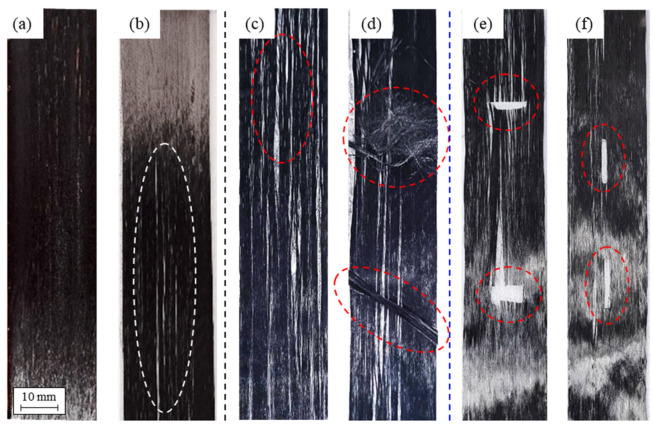
Classification of carbon fiber (CF) tows based on damage status: (**a**) normal; (**b**) fibers within 1 mm in width and intra-fiber defects present (assessed as normal); (**c**) fibers larger than 1 mm in width and inter-fiber defects present (poor); (**d**) dust, contact by foreign objects on the fiber tow, and separation of fiber and inter-fiber interfaces; (**e**) fiber fractured in the horizontal direction; and (**f**) fiber fractured in the vertical direction.

**Figure 10 polymers-15-04182-f010:**
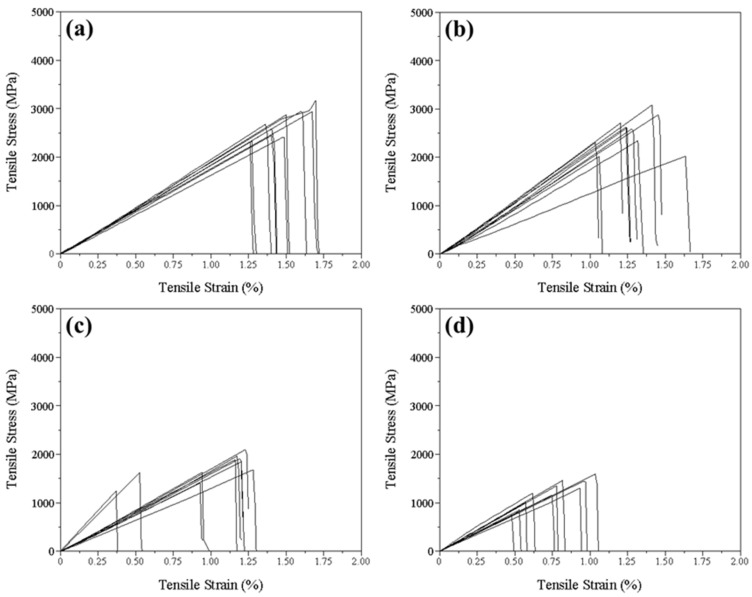
Tensile stress of single-fiber depending on various individual fibers after ECT: (**a**) normal, (**b**) uneven state, (**c**) foreign substances, and (**d**) cut single-fiber.

## Data Availability

The data that supports the findings of this study are available within the article.

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
