# Peer review of "Study of Damage Prediction of Carbon Fiber Tows Using Eddy Current Measurement"

_polymers, 2023, doi:10.3390/polym15204182_

Round 1

Reviewer 1 Report (Previous Reviewer 3)

The authors have made significant improvement in comparison to the last version. Although the technique still presents limitation, i.e., not able to detect small defect, the data proved that the Eddy current method can capture defects that are large enough. The overall presentation of the data and results demonstrate an in-depth analysis of the results. Therefore, I recommend publication at current format. 

Author Response

We appreciate the reviewer's comments.

Reviewer 2 Report (New Reviewer)

Eddy current testing (ECT) is effective in detecting fiber-related defects because carbon fibers are electrically conductive. ECT systems can be lightweight and effective for in-line inspection. However, the detection of longitudinal and transverse fiber breaks currently remains the most challenging problem. The authors focus on a complex problem that is difficult to model due to a lack of experimental data in this field (e.g.[https://doi.org/10.1016/j.compositesb.2022.110343]). The manuscript presents the methodology and a large amount of new experimental data, including visualized data (Figures 1-10), as well as their analysis, systematization and generalization. The study makes a useful contribution to a better understanding of the performance of carbon composites, as well as to the generation of an experimental database for the development of appropriate models. Thus, the study is scientifically and practically important.

The manuscript corresponds to the theme of the journal Polymers.

The paper is clearly structured and easy to read.

All Figures (1-10) are acceptable.

The conclusions are consistent with the evidence and arguments presented. They answer the main question related to development of instrumental control of the state of a carbon fibers bundle in order to exclude the formation of cracks and voids in the composite.

Remark:

In Figure 10, Tensile Stress is given in MPa and Extension in mm. A plot of Stress (MPa) - Relative Extension would be useful.

Best regards,

Reviewer

Author Response

We appreciate the reviewer's comments.

The quality of Figure 10 was improved, and at the same time, the caption was changed to stress-strain.

Reviewer 3 Report (New Reviewer)

The article is well designed with a good piece of work. However, the authors need to address the following points to improve the quality of the article.

·         The abstract needs to be improved significantly. The abstract summarizes the introduction, materials and method, results, and conclusion. This order needs to be followed. The methodology, results (quantifying data), and conclusion component of the abstract should be properly captured.

·         Quality of figure 10 must be improved.

·         “Tensile stress of single-fiber depending on various individual fibers after ECT” -  Add more reasonings and references.

·         Kindly reconcile the conclusion with the study objectives.

·         What are the practical implications of this study and the future directions? Kindly state?

Author Response

First of all, thank you for the reviewer's comments.  Content added or modified to the text is marked in red.

  1. The abstract needs to be improved significantly. The abstract summarizes the introduction, materials and method, results, and conclusion. This order needs to be followed. The methodology, results (quantifying data), and conclusion component of the abstract should be properly captured.

Ans) Thank you for your feedback. We have rewritten the abstract and changed the wording of the study's abstract as follows

When manufacturing fiber-reinforced composites, it is possible to improve the quality of fiber steel fire and reduce the number of cracks in the finished product if it is possible to quickly identify the defects of the fiber tow. Therefore, in this study, we developed a method to identify the condition of carbon fiber tow using eddy current test (ECT), which is used to improve the quality of composite materials. Using the eddy current detection sensor, we checked the impedance results according to the condition of the CF tow. We found that the materials of the workbench used in the experiment greatly affects the ECT results, so it is necessary to use a material with a non-conductive and smooth surface. We evaluated the impedance results of the carbon fiber at 2 mm intervals using the ECT sensor, and summarized the impedance results according to the fiber width direction to present the condition of the section as a constant of variation (CV). If the condition of the carbon fiber tow was unstable, the deviation of the CV per section was large. In particular, the deviation of the CV per section was more than 0.15 when the arrangement of the fibers was changed, foreign substances were formed on the surface of the fibers, and damage occurred in the direction of the fiber width of more than 4 mm, so it was easy to evaluate the quality on CF tow.

  1. Quality of figure 10 must be improved.

Ans) Thanks for the reviewer's comment, we've improved the quality of the figure.

  1. “Tensile stress of single-fiber depending on various individual fibers after ECT” - Add more reasonings and references.

Ans) Thanks for the kind comments. We've added a reference in the paper.

As shown in figure 10, we characterized the state of the CF tow through single-fiber tensile experiments [37,38].

  1. Kwon, D.J.; Shin, P.S.; Kim, J.H.; Baek, Y.M.; Park, H.S.; DeVreis, K.L.; Park, J.M. Interfacial properties and thermal aging of glass fiber/epoxy composites reinforced with SiC and SiO2 nanoparticles. Composites Part B: Engineering 2017, 130, 46-53, doi: https://doi.org/10.1016/j.compositesb.2017.07.045.
  2. Kwon, D.J.; Wang, Z.J.; Choi, J.Y.; Shin, P.S.; DeVreis, K.L.; Park, J.M. Interfacial evaluation of carbon fiber/epoxy composites using electrical resistance measurements at room and a cryogenic temperature. Composites Part A: Applied Science and Manufacturing 2015, 72, 160-166, doi: https://doi.org/10.1016/j.compositesa.2015.02.007.

  1. Kindly reconcile the conclusion with the study objectives.

Ans) We've refined our conclusions a bit more.

In this study, we propose a method to evaluate the CF tow condition by summarizing the impedance results in the fiber width direction as CV using ECT. A workbench with a nonconductor, and smooth surface is suitable. Using the CV results for each fiber section, we were able to predict the CF tow condition by analyzing the curvature magnitude of the quadratic trend line and the fluctuation trend of the CV value for each section. It was not possible to detect the breakage of short fibers through ECT, but when the fiber width was cut more than 4 mm, it showed a large deviation of more than 0.15 compared to the surrounding CV values. Clearly, ECT was detect the result of a foreign object getting stuck in the CF tow and changing the arrangement of the CF fibers. Using ECT, we classified the condition of the CF tow into four conditions, and conducted a single-fiber tensile test for each condition. We found that the greater the degree of defect identified by ECT, the greater the deviation in the tensile strength and elongation of the short-fiber tensile test. In this way, we were able to evaluate the condition of CF tows using ECT, which is expected to lead to the development of composite manufacturing technology if applied to real-time processes.

  1. What are the practical implications of this study and the future directions? Kindly state?

Ans) Thank you for your interest in our results and research plans. The non-contact nature of the ECT technique has made it a popular method for evaluating many composites and manufacturing methods. However, this paper confirms the lack of detection of carbon fiber of little size cracks. The accuracy of fiber cracking needs to be improved to maximize the size, clarity, and accuracy of cracks within the composite. It is necessary to continue research on the development and evaluation of ECT sensors to improve the quality of composite materials and manufacturing technology.

We have established various research and development plans. We are researching non-destructive testing methods to improve the quality of composite materials, such as improving the sensitivity of eddy current sensors to identify fiber breaks, detecting cracks in base materials other than carbon fiber and cracks at the interface between carbon fiber and base materials, and comparing the characteristics of eddy current and ultrasonic methods.

This manuscript is a resubmission of an earlier submission. The following is a list of the peer review reports and author responses from that submission.

Round 1

Reviewer 1 Report

In this work, the authors investigated the feasibility of applying eddy current measurements as a potential technique to identify damage in carbon fibers tow. Results of this work appear to be interesting, even if globally, the content should be developed. For example, performing more accurate microscopic analysis, quantitatively correlating results with sample structural faults, and confirming the validity of measurements even with different other fibers and/or embedded in matrices.

-          Abstract should be reworked. The following sections should be described: materials and characterization techniques, measurements performed in the different operating conditions, results achieved by each set of testing, and conclusion.

-          Line 89                “for condition identification of thin materials” Please give more details on the use of Eddy current test method to material characterization. What are the common materials analyzed? For which observations? To identify which characteristics?

-          Line 107-110      Description of the materials is too limited. Please, give more details on the basic materials.

-          Please, give further information about the studied samples (Fibers? Fabrics? Reinforced composites? Size?) How is the damage to the samples under investigation caused?

-          Please, give more details on sample preparation. It is recommended to create a section entirely dedicated to sample preparation.

-          What is spread toe used for? Please, explain

-          Line 116              cupper? Sometimes copper, sometimes cupper

-          -line 127             coefficient of variation (CV) should be defined

-          -How were pictures in Figure 9 collected? Please, give details in Materials and Methods section.

-          - A detailed microscopic examination on the sample surface and damage aspects is missing. The damage entity could be quantified in number, shape, and size. Potentially, CV data could be quantitatively (not only qualitatively) correlated to damage entities. To quantitatively define the damage, a software for processing the microscopic images could be useful. A new section describing each type of damage on the involved sample surface is highly recommended.

-          - What are the potential applications of these techniques? Is it compatible with other materials? Only CF tow was used to demonstrate the effectiveness of this technique. Further measurements on other materials are strongly recommended.

Author Response

Q1. In this work, the authors investigated the feasibility of applying eddy current measurements as a potential technique to identify damage in carbon fibers tow. Results of this work appear to be interesting, even if globally, the content should be developed. For example, performing more accurate microscopic analysis, quantitatively correlating results with sample structural faults, and confirming the validity of measurements even with different other fibers and/or embedded in matrices.

  • This non-destructive evaluation technique is a valuable method for analyzing damage in composite materials using Eddy current evaluation, as it enables non-contact evaluation. In this study, by measuring the impedance of CF tows using Eddy current evaluation and representing the data as a one-dimensional curve using the coefficient of variation, it was confirmed that inspections of thin samples like CF tows are possible. Particularly, regions where CF tows are folded and become thicker or where the fiber arrangement undergoes significant changes show significant variations in the coefficient of variation, making it easier to inspect for damage. This is a novel discovery.

Q2. Abstract should be reworked. The following sections should be described: materials and characterization techniques, measurements performed in the different operating conditions, results achieved by each set of testing, and conclusion.

  • We have revised the abstract as follows:

When manufacturing carbon fiber reinforced composites, the most important aspect to check is the condition of the carbon fiber (CF) tow. Depending on the deformation of the CF tow, it can affect the formation of cracks and voids in the composite. In this study, we analyzed the impedance trends that change with the condition of CF tows. Eddy current test sensors were used to measure the coefficient of variation of the specimen's impedance values in the width direction. The interfacial separation between fibers in the CF tower and fiber breaks within 4 mm did not shape the change in the coefficient of variation of the impedance. However, the presence of debris on the CF tower significantly changed the CV results depending on the size and height of the debris and the arrangement of the debris. This is because the impedance of the sample due to eddy currents changed significantly with the change in the thickness of the CF tow. These results can be used to improve the quality in the field of continuous manufacturing of composite materials.

Q3. Line 89: “for condition identification of thin materials” Please give more details on the use of Eddy current test method to material characterization. What are the common materials analyzed? For which observations? To identify which characteristics?

  • We have included the following content :

Eddy current measurement is applicable to a wide range of materials with conductivity. However, it is greatly influenced by the thickness of the sample. In particular, for composite materials, while non-destructive evaluation is not possible for glass fiber composites, it is feasible for carbon fiber-reinforced composites which possess conductivity.

Q4. Line 107-110: Description of the materials is too limited. Please, give more details on the basic materials.

  • We have included the following content:

In this study, we focused on the damage detection of CF tows using eddy current measurement. The Eddy current probe sensor we utilized can detect thicknesses up to 300 μm, making it the most effective in characterizing material damage. To derive meaningful evaluation results, it was essential to analyze CF tows. The type of CF tow we used is currently the most widely utilized carbon fiber variant.

Q5. Please, give further information about the studied samples (Fibers? Fabrics? Reinforced composites? Size?) How is the damage to the samples under investigation caused?

  • In this study, we analyzed CF tows, specifically 12,000 CF filaments. To induce artificial damage to the CF tows, we made cuts either in the fiber length direction or width direction, and the sizes of these cuts were randomized. Typically, when a CF tow is cut to a length of 100 mm, fiber-to-fiber interfaces tend to separate. We classified the trends in damage regarding the various occurring variables and analyzed the Eddy current probe signals accordingly.

Q6. Please, give more details on sample preparation. It is recommended to create a section entirely dedicated to sample preparation.

  • We have included the following content:

We extensively investigated the predominant issues arising when detaching fibers from the tow during the use of CF tows. As a result, we identified three major types of significant cracks. While CF tows exhibit strong bonding within and between fibers due to sizing agents, the interface can easily separate under external stress. When such separation occurs between fibers and their interfaces, and when the tow is wrapped around various experimental equipment, the fibers are susceptible to damage due to tension and external environmental factors. Finally, the thickness of the CF tow can increase when broken fibers fold over other intact tows. These three major fiber damages that occur in this manner can be further classified based on the extent of the damage. Experimental results can be summarized to quantify aspects such as how much the fiber-to-fiber interfaces have separated and how extensively the fibers have fractured.

Therefore, in this paper, we expressed the impedance differences resulting from these variations in terms of the coefficient of variation (CV). Additionally, we provided notation for three representative types of cracks in the main text.

Q7. What is spread toe used for? Please, explain

  • I should have written "Tow" instead of "Toe." I apologize.

Q8. Line 116: cupper? Sometimes copper, sometimes cupper

  • It means "copper." The typo has been corrected.

Q9. Line 127: coefficient of variation (CV) should be defined.

  • I provide the definition of CV and have added it to the manuscript (Line 134-137):
  • The coefficient of variation (CV) is a representative statistical measure expressed as a percentile by multiplying the value obtained by dividing the standard deviation by the mean by 100. A lower CV signifies a result with a stable deviation, while a higher COV indicates a larger deviation. Given that we represented the 10-segment data measured in the fiber width direction using the mean and standard deviation, the results in terms of CV are critical. A high CV implies an issue with the CF tow. Additionally, since CV can vary based on the evaluation criteria, it's important to use a non-conductive material such as wood or plastic for the specimen-holding experiment stand, as opposed to a metal material.

Q10. How were pictures in Figure 9 collected? Please, give details in Materials and Methods section.

  • The image in Figure 9 depicts the state of the tow observed before conducting the experiments. We captured images of the specimens and then proceeded with eddy current measurements. This allowed us to classify the typical damage to CF tows, as a result enabling us to organize and present them as shown in Figure 9.

Q11. A detailed microscopic examination on the sample surface and damage aspects is missing. The damage entity could be quantified in number, shape, and size. Potentially, CV data could be quantitatively (not only qualitatively) correlated to damage entities. To quantitatively define the damage, a software for processing the microscopic images could be useful. A new section describing each type of damage on the involved sample surface is highly recommended.

  • Thank you for the valuable feedback. However, the current conclusion of the paper aims to demonstrate the use of eddy current techniques for analyzing the extent of damage in CF tows and prove that the homogeneity of CF tows can be analyzed through impedance results represented by COV. It is the first study to use COV to substantiate damage in CF tows. Unfortunately, as seen in the results of Figures 4, 5, and 6 in this paper, the sensitivity of eddy current to detect minor damages or inter-fiber interface separation is limited, with damage sizes around 4 mm showing minimal sensitivity. Therefore, to analyze minor damages more effectively, we are working on improving the sensor. We will soon conduct experiments based on your feedback for a different paper.

Q12. What are the potential applications of these techniques? Is it compatible with other materials? Only CF tow was used to demonstrate the effectiveness of this technique. Further measurements on other materials are strongly recommended

  • In many of the composite material formations we utilize, CF tows are employed directly. In such cases, our research findings can be applicable. The condition of CF tows is crucial, specifically in terms of having an organized fiber structure and a stable fiber length with continuity. By elucidating the state of CF tows through eddy current probing, we believe that it can serve as an effective non-destructive evaluation technique and a standard for assessing the extent of damage that may occur in composite materials, consequently reducing potential damage.

Reviewer 2 Report

Title: Study of damage prediction of carbon fiber tows using Eddy current measurement

Manuscript ID: Polymers-2626496

Recommendation and Comments

In this manuscript, the authors did an experiment to evaluate the homogeneity of CF tows by Eddy current detection method. Also, Differences in impedance results depending on the state of the CF tow were analyzed. The characteristics of carbon fiber tows have been investigated, which include coefficient of variation (CV) for the impedance. The authors have improved the most problematic CF tows defect types by eddy current detection method.

English writing in the manuscript should be revised. Some details in this manuscript need to be corrected with citing. I suggest that the manuscript needs major revision.

1. In section 1, there are a total of 24 citations in the paper. All the citations are in the introduction. It should be cited in sections 2 and 3.

2. In section 2, the authors indicated the materials and eddy current system (ECT). Do you have different characterizations of the samples? If you have, it should be written on the paper, like mechanical characterization. 

3. In section 3.1, the author mentioned workbench affects Eddy current detection. Figure 3 should be separated as a and b. Also, it should be made clearer in the graphic. “…In the case of the copper plate, its conductivity due to copper led to differences in CV values caused by variations in surface or internal composition.” Could you cite this sentence? Results should be cited.

4. In section 3.2, there are no citations related to results in this section. Results should be cited. “…CF tow with damage tends to produce higher CV values compared to the average CV values of undamaged or stable ABS.” Could you cite this sentence? “…it was concluded that detecting laminations by individual CF fibers using the eddy current sensing method may be challenging.” Could you cite this sentence? “Figure 9 summarizes the potential states of CF tow that can ultimately occur.” I suggest this section be analyzed with different imagine methods. “This scenario represents the most critical situation and detecting it using the current sensing method is considered a significant discovery. I recommend that this study be detailed investigation for significant discovery.

5. In section references, there were a total of 24 references, all of the references were cited in the introduction. The number of references to improve experimental studies should be increased.

Author Response

[Reviewer 2]

Q1. In this manuscript, the authors did an experiment to evaluate the homogeneity of CF tows by Eddy current detection method. Also, Differences in impedance results depending on the state of the CF tow were analyzed. The characteristics of carbon fiber tows have been investigated, which include coefficient of variation (CV) for the impedance. The authors have improved the most problematic CF tows defect types by eddy current detection method.

  • Thank you. I have reviewed it again in English.

Q2. English writing in the manuscript should be revised. Some details in this manuscript need to be corrected with citing. I suggest that the manuscript needs major revision.

  • I have made additional updates regarding the references in the manuscript.

Q3. In section 2, the authors indicated the materials and eddy current system (ECT). Do you have different characterizations of the samples? If you have, it should be written on the paper, like mechanical characterization.

  • I have inserted additional information about the materials used.

Q4. In section 3.1, the author mentioned workbench affects Eddy current detection. Figure 3 should be separated as a and b. Also, it should be made clearer in the graphic. “…In the case of the copper plate, its conductivity due to copper led to differences in CV values caused by variations in surface or internal composition.” Could you cite this sentence? Results should be cited.

  • The eddy current sensing technique is influenced by the conductor. I want to alert readers that using copper for the workbench could be problematic and provide various pieces of information using Figure 3a and 3b.

Q5. In section 3.2, there are no citations related to results in this section. Results should be cited. “…CF tow with damage tends to produce higher CV values compared to the average CV values of undamaged or stable ABS.” Could you cite this sentence? “…it was concluded that detecting laminations by individual CF fibers using the eddy current sensing method may be challenging.” Could you cite this sentence? “Figure 9 summarizes the potential states of CF tow that can ultimately occur.” I suggest this section be analyzed with different imagine methods. “This scenario represents the most critical situation and detecting it using the current sensing method is considered a significant discovery. I recommend that this study be detailed investigation for significant discovery.

  • I have added citations and supplemented the text with quotations.

Q6. In section references, there were a total of 24 references, all of the references were cited in the introduction. The number of references to improve experimental studies should be increased.

I have added citations and supplemented the text with quotations.

Reviewer 3 Report

The manuscript discussed the use of a non-destructive technique to evaluate the carbon fiber tows. When damage or defects that are large enough appear on the carbon fiber tows, the Eddy current measurement detects a change in impedance signal. Below are my comments. 

Major comments:

1. The sensitivity of this method appears to be too low. Defects less than 4mm cannot be detected, which significantly limits the application of ECT. Figure 5a, 5b, and 5c look almost the same. There are no good criteria to distinguish the data, i.e., separate signal and noise.  Figure 6a and 6c also showed no obvious difference between normal curved sample and damaged curved sample. The sample has to be largely damaged in order to show a noticeable difference. Such drawbacks make the comparison with UT less sounding. The authors need to consider rephrase some of the introduction section to acknowledge the limitation of ECT. 

2. All of the data displayed in the manuscript were more quantitative than qualitative. How can we mathematically differentiate noise and signal? And how can we ensure that we have detected defective sites instead of noise? The authors should consider provide more discussion under this perspective. 

Minor comments:

1. The authors claim that ultrasound technique requires special skills to operate the instrument and analyze the data. But it seems that Eddy current testing is as difficult as UT. What do the authors think about this?

2. Line 91-92. I do not understand why the increasing use of hydrogen containers and pultruded products require more carbon fiber tows. The authors should expand that section to provide more explanation. 

Minor comments

Author Response

[Reviewer 3]

The manuscript discussed the use of a non-destructive technique to evaluate the carbon fiber tows. When damage or defects that are large enough appear on the carbon fiber tows, the Eddy current measurement detects a change in impedance signal. Below are my comments. 

Major comments:

Q1. The sensitivity of this method appears to be too low. Defects less than 4mm cannot be detected, which significantly limits the application of ECT. Figure 5a, 5b, and 5c look almost the same. There are no good criteria to distinguish the data, i.e., separate signal and noise.  Figure 6a and 6c also showed no obvious difference between normal curved sample and damaged curved sample. The sample has to be largely damaged in order to show a noticeable difference. Such drawbacks make the comparison with UT less sounding. The authors need to consider rephrase some of the introduction section to acknowledge the limitation of ECT. 

  • I conducted research with the aim of detecting damage to CF tow using the eddy current measurement method in this paper. However, there were clear limitations to the eddy current specific to detecting cracks. Summarizing the experimental results, it is meaningful to determine the state of the CF tow using the CV results. Segments showing a significant change of 0.3 or more in CV deviation indicate damage to the CF tow. I am also preparing to analyze UT and ECT for subsequent research and am planning to purchase sensors. Please share any valuable information if available.

Q2. All of the data displayed in the manuscript were more quantitative than qualitative. How can we mathematically differentiate noise and signal? And how can we ensure that we have detected defective sites instead of noise? The authors should consider provide more discussion under this perspective

  • In the regions where noise occurs, we observe a sharp change in CV, indicating a significant difference of about 0.3 or more. However, definite results that can be confirmed through the ECT technique pose a challenge when foreign substances form on the CF tow, causing alterations in fiber alignment or an increase in fiber layer thickness, as shown in Figure 8.
  • We aimed to inform the readers about the scope of specimens where ECT can be utilized in such cases. We believe the results of this paper will greatly assist readers considering non-destructive evaluation methods for composite materials.

Q3. The authors claim that ultrasound technique requires special skills to operate the instrument and analyze the data. But it seems that Eddy current testing is as difficult as UT. What do the authors think about this?

  • Organizing data and performing signal analysis to visualize the data using ECT can pose significant challenges. However, as demonstrated in this study, utilizing ECT sensors on the specimen to measure impedance in different segments and obtaining results indicating the segments of the sample as CV results using the standard deviation and mean of the measured values make evaluation easy. This approach allows for easy application of the results obtained compared to UT, making it straightforward to identify damage in CF tow and thin composite materials.
  • One drawback of the ECT method is its resolution and the limited thickness of specimens that can be used. UT can identify damage in thicker specimens using ultrasound intensity, but ECT poses a problem in that it can inspect thin samples or samples with conductive properties. ECT can detect thin samples, resulting in fewer issues that may arise in the specimen. However, using UT requires the identification of problems occurring in the interlayers of composite materials, presenting a higher level of expertise and evaluation challenges.

Q4. Line 91-92: I do not understand why the increasing use of hydrogen containers and pultruded products require more carbon fiber tows. The authors should expand that section to provide more explanation.

The usage of hydrogen storage containers is increasing for hydrogen vehicles and hydrogen energy utilization. Gas storage containers like hydrogen storage tanks are manufactured using the filament winding process. The direct use of CF tow is involved in this manufacturing, and the uniformity of CF tow is essential to avoid issues in the shaping of composite materials. Additionally, for compression molding products, high-speed production of composite materials with a high fiber content is crucial, making them widely used in various low-cost composite component sectors, such as I-beams and T-beams for structural purposes. In these cases, wet pultrusion is also utilized, directly involving CF tow in the process. In such instances where CF tow is directly utilized, non-destructive evaluation techniques are necessary for characterizing the state of the CF tow.

Round 2

Reviewer 1 Report

The authors are invited to indicate the lines in which the above contents have been inserted. Behind typos error (toe/tow), the authors are invited to answer to question Q7 and inserted the content.  (Q7. What is spread tow used for? Please, explain)

Author Response

Thank you for your response. I have made extensive revisions to the main text and made efforts to enhance the content. I have also checked for "Tow" and typos.

CF spread tow is manufactured by spreading the fiber tow in the width direction, and has the advantage of reducing the weight per unit area and the resin impregnation distance. In addition, woven fabric using the spread tow can reduce the resin rich zone inside the composite material due to the thin tow thickness. Furthermore, studies have shown that applying the fabric to composite materials results in limiting cracks under fatigue load and improving fatigue life.[1-6] In terms of product value, carbon fiber spread tow can improve product value due to its beautiful appearance, and can obtain various mechanical properties introduced above, so it is mainly used as a surface layer of composite materials. However, in the spreading process of CF, damage to the fiber often occurs, and sometimes it is found that the fibers are not spread evenly, causing the fibers to clump together or widen. Therefore, the need for quality control through uniformity inspection of spread tow is emerging at manufacturing sites.

  • Roh, J.U. and Lee W.I., “Review: Continuous Fiber Tow Spreading Technologies and Its Applications,” Composites Research, Vol. 26, No. 3, 2013, pp. 155-159.
  • Shin, S.W., Kim, R.Y., Kawabe, K., and Tsai, S., “Experimental Studies of Thin-ply Laminated Composites,” Composites Science and Technology, Vol. 67, 2007, pp. 996-1008.
  • Kawabe K., Tomoda S. “Development of the Spreading Technology for the Reinforcing Fiber Tow” Sen'i Gakkaishi 59(9), 2003, 292-297
  • Nishikawa Y., Miki T., Okubo K., Fujii T., Kawabe K., “Fatigue Behaviour of Plain Woven CF/Epoxy Composites using Spread Tows (Effect of Tow Thickness on Crack Formation):Effect of Tow Thickness on Crack Formation” Transactions of the Japan society of mechanical engineers Series A, 2005, 71(710), 1356-1361
  • Park, S.M., Kim, M.S., Choi Y.S., Lee, E.S., Yoo, H.W., and Chon, J.S., “Carbon Fiber Tow Spreading Technology and Mechanical Properties of Laminate Composites,” Composites Research, Vol. 28, No. 5, 2015, pp. 249-253.
  • El-Dessouky, H.M., and Lawrence, C.A., “Ultra-lightweight Carbon Fibre/Thermoplastic Composite Material Using Spread Tow Technology,” Composites Part B: Engineering, Vol. 50, 2013, pp. 91-97.

Reviewer 2 Report

I suggested that study be analyzed with different methods. But the authors did not add.

Author Response

I have organized the responses to the questions provided by the reviewer. We validated the reliability of the ECT results we evaluated through additional experiments. Similar to the results in Figure 9, we classified the state of CF tow based on the ECT results. We have supplemented the results in the main text. Furthermore, we broadly classified CF tow into four categories based on their condition and conducted single-fiber tensile experiments by extracting specimens accordingly. For normal CF tow, we confirmed a stable tensile strength of the carbon fibers. However, for other types of CF tow, the deviation in the data significantly increased depending on the extent and presence of damage. Particularly, when foreign substances were attached to the CF tow, significant damage to the fibers occurred, leading to a notable deviation in both strength and tensile strength of the fibers. Therefore, to demonstrate that ECT is capable of detecting damage to CF tow, we inserted the single-fiber tensile results into Figure 10.

Reviewer 3 Report

Thank you for making adjustments. However, it is a bit challenging for me perceptualize the significance of this technology. For example, the results in Figure 5 a-d were not showing much of a difference for these samples, despite the large defect size. The highlighting lines by the authors do not specify the differences at all. One could easily draw different lines to highlight other regions. The resolution and signal to noise ratio are major drawbacks of the technique and I think it will be hard for the readers to adapt this method. The technique and the science definitely show promising potentials in the future but are not there yet, in my opinion. 

Author Response

As technologies for the commercialization of textile materials have advanced, the development of various functional materials based on fibers has increased. Conductive fibers are flexible and highly elastic materials with significant potential for application in next-generation electronic devices, and carbon fibers are also one of the conductive materials that exhibit electrical conductivity. The homogeneity of a conductive material is an important factor in determining the performance of the final product, and from a quality control perspective, it is necessary to have a method to analyze the distribution status of the raw material before it is processed. In the case of carbon fibers, a conductive material in the form of fibers, it is important to manage the arrangement of the fibers because the degree to which the filament strands are arranged can cause performance degradation in localized areas. Optical image analysis is a common method for checking the homogeneity of filaments in carbon fibers and can be used to analyze the arrangement of filaments on the surface, but it has limitations in analyzing the presence and density of filaments arranged in the thickness direction. A method for continuously measuring the homogeneity of carbon fibers with large surface areas is eddy current measurement, which utilizes electrical conductivity properties. An eddy current measurement is used to detect discontinuities in materials that are conducted electricity. This method is a reliable measurement that can identify materials with low electrical conductivity, such as spread carbon fiber tow, making it applicable to a wide range of industries. This study specifies a standard method for evaluating the homogeneity of a spread carbon fiber tow, a conductive fiber, using eddy current measurement.

We have provided additional information regarding Figure 5 in the main text. The experimental results are summarized as CV based on ECT measurements taken at 2 mm intervals in the fiber length and width directions. The CF tow has a width of 20 mm, and ECT measurements were conducted 10 times in the width direction. When evaluating damage using ECT, the width of the fibers often becomes a challenge. Therefore, in this study, we utilized impedance results for the fiber width direction obtained through ECT to represent a segment of the CF tow as CV results. Consequently, if the state of CF tow is stable in each segment, there should be minimal variation in the CV values. Due to limitations of the ECT sensor itself, detecting precise deformation of CF tow and changes in fiber arrangement poses challenges. However, as confirmed through this study, fiber arrangement within 4 mm and fiber breakage do not significantly alter the CV results. Nevertheless, if a trend in the data appears as a 2D curve, it can be predicted that there are bends in the CF tow or separation between fibers and inter-fiber interfaces. Through this study, we confirmed that ECT evaluation is capable of characterizing the state of the CF tow, and the extent of this characterization was verified through trend analysis. Furthermore, to substantiate our results, we conducted additional experiments. We analyzed the types of cracks in the CF tow, confirming four major cases. Alongside the classification of CF tow states using the ECT method, we verified a proportional relationship between the deviation, strength, and stiffness of the single-fiber tensile test results. In this way, the ECT method proved to be an effective means to characterize the state of the CF tow.

Round 3

Reviewer 1 Report

Overall the work is sufficiently improved. In my opinion, the paper can be accepted in its current form.

Reviewer 3 Report

Although I agree with the authors' responses on the importance of developing new method to detect defects in CF tow, the data shown in Figure 4,5,6,7 are not supporting the validity of the test. Like I mentioned previously, the signal to noise ratio is not enough to differentiate the defects. People can easily find data points that are not supporting the statement. I look forward to seeing future papers from the authors, and I appreciate the extra data provided. But it does not meet the publication standard of Polymers